# Quantifying the impact of public omics data

Yasset Perez-Riverol [1], Andrey Zorin[1], Gaurhari Dass [1], Manh-Tu Vu[1], Pan Xu[2], Mihai Glont[1], Juan Antonio Vizcaíno [1], Andrew F. Jarnuczak [1], Robert Petryszak[1], Peipei Ping[3,4] & Henning Hermjakob [1,2]

The amount of omics data in the public domain is increasing every year. Modern science has become a data-intensive discipline. Innovative solutions for data management, data sharing, and for discovering novel datasets are therefore increasingly required. In 2016, we released the first version of the Omics Discovery Index (OmicsDI) as a light-weight system to aggregate datasets across multiple public omics data resources. OmicsDI aggregates genomics, transcriptomics, proteomics, metabolomics and multiomics datasets, as well as computational models of biological processes. Here, we propose a set of novel metrics to quantify the attention and impact of biomedical datasets. A complete framework (now integrated into OmicsDI) has been implemented in order to provide and evaluate those metrics. Finally, we propose a set of recommendations for authors, journals and data resources to promote an optimal quantification of the impact of datasets.

[1] European Molecular Biology Laboratory, EMBL-European Bioinformatics Institute (EMBL-EBI), Cambridge CB10 1SD, UK. [2] State Key Laboratory of Proteomics, Beijing Proteome Research Center, Beijing Institute of Lifeomics, National Center for Protein Sciences (The PHOENIX Center, Beijing), 102206 Beijing, China. [3] Department of Physiology, Division of Cardiology, David Geffen School of Medicine at UCLA, University of California, Los Angeles 90095 CA, USA. [4] Department of Medicine, Division of Cardiology, David Geffen School of Medicine at UCLA, University of California, Los Angeles 90095 CA, USA. Correspondence and requests for materials should be addressed to Y.P.-R. (email: yperez@ebi.ac.uk)

Public availability of datasets is growing in all disciplines, because it is considered to be a good scientific practice (e.g. to enable reproducibility) and/or it is mandated by funding agencies and scientific journals[1,2]. Science is now a data intensive discipline and therefore, new and innovative ways for data management, data sharing and for discovering novel datasets are increasingly required[3,4]. However, as data volumes grow, quantifying data impact becomes more and more important. In this context, the Findable, Accessible, Interoperable, Reusable (FAIR) principles have been developed to promote good scientific practises for scientific data and data resources[5]. In fact, recently, several resources[1,2,6] have been created to facilitate the Findability (F) and Accessibility (A) of biomedical datasets. These principles put a specific emphasis on enhancing the ability of both individuals and software to discover and re-use digital objects in an automated fashion throughout their entire life cycle[5]. While data resources typically assign an equal relevance to all datasets (e.g. as results of a query), the usage patterns of the data can vary enormously, similarly to other "research products" such as publications. How do we know which datasets are getting more attention? More generally, how can we quantify the scientific impact of datasets?

Recently, several authors[7–9] and resources[10] pointed out the importance of evaluating the impact of each research product, including datasets. Reporting scientific impact is indeed increasingly relevant for individuals, but also reporting aggregated information has become essential for research groups, scientific consortia, institutions or for public data resources among others, in order to assess the level of importance, excellence and relevance of their work. This is a key piece of information for funding agencies, which is used routinely to prioritise the projects and resources they fund. However, most of the efforts nowadays focus on the evaluation and quantification of the impact of publications as the main artefact. For instance, in 2013, the "altmetrics" team proposed a set of 'alternative' metrics to trace research products with special focus on publications[10]. Specific tools and services have been built since to aggregate "altmetrics", including for instance counts of mentions of a given publication in blog posts, tweets and articles in mainstream media. The altmetrics attention score is widely used by the research community nowadays (e.g. by multiple scientific journals), as a measure of scientific influence of manuscripts. However, adequate tracking and recognition of datasets has been limited so far for multiple reasons: (i) the relatively low number of publications citing datasets instead of their corresponding publications; (ii) the lack of services that store and index datasets from heterogeneous origins; and (iii) the absence of widely used metrics that enable the quantification of their impact. Some attempts have been made to improve the situation, by introducing data object identifiers (DOIs) directly associated to datasets[11].

In 2016, we released the first version of the Omics Discovery Index (OmicsDI—https://www.omicsdi.org) as a light-weight system to aggregate datasets across multiple public omics data resources. OmicsDI aggregates genomics, transcriptomics, proteomics, metabolomics and multiomics datasets, as well as computational models of biological processes[1]. The OmicsDI web interface and Application Programming Interface (API) provide different views and search capabilities on the indexed datasets. Datasets can be searched and filtered based on different types of technical and biological annotations (e.g. species, tissues, diseases, etc.), year of publication and the original data repository where they are stored, among others. At the time of writing (March 2019), OmicsDI stores just over 454,200 datasets from 16 different public data resources (https://www.omicsdi.org/database). The split per omics technology is as follows: transcriptomics (125,891 datasets), genomics (309,961), proteomics (12,362),

metabolomics (2411), multiomics (6578) and biological models (8651). Here, we propose a set of novel metrics to quantify the impact of biomedical datasets. A complete framework (now integrated in OmicsDI) has been implemented in order to provide and evaluate those metrics. Finally, we propose a set of recommendations for authors, journals and data resources to promote an optimal quantification of the impact of datasets.

## Results

**Omics data reanalysis and citations.** By March 2019, the number of datasets with at least one reanalysis, one citation, one download, one view and that contained connections in knowledgebases was 12,162, 58,054, 66,418, 163,431 and 469,015, respectively (Table 1). The reanalysis metric quantifies how many times one dataset has been re-used (re-analysed) and the result deposited in the same or in another resource. We classify reanalyses in two different categories: (i) reanalyses performed by independent groups (Independent Lab Reanalyses) or reanalyses performed systematically by resources such as PeptideAtlas or Expression Atlas (Resource Reanalyses). On average, each reanalysed dataset is reanalysed 2.3 times. However, each omics type has a different pattern: proteomics (5.90), transcriptomics (1.31), multiomics (2.07), genomics (1.26) and models (30.08).

Frequently, dataset re-use is a hierarchical process, where one dataset is reanalysed subsequently multiple times. Figure 1a presents a reanalysis network for the model BIOMD0000000055, starting from 2006 (release year) to 2015. A different pattern is illustrated in Fig. 1b, where BIOMD0000000286 is derived from multiple source models. BioModels curates and annotates for each deposited model, the corresponding model from which it is derived (if applicable). Figure 1c shows the reanalysis network of the PRIDE dataset PXD000561 (https://www.omicsdi.org/dataset/pride/PXD000561) (75); one of the "drafts of the human proteome". This dataset and the PXD000865 have supported the annotation of millions of peptides and proteins evidences, enabling the large-scale annotation of the human proteome[12] and have been reanalysed by multiple databases including the proteomics resources PeptideAtlas and GPMDB[13].

Interestingly, the distribution of the elapsed time between the year of publication of the original datasets and publication of the reanalyses shows that most of the datasets are reanalysed within the first 5 years after publication (Fig. 2a). After 10 years of publication, still datasets are often reused in public databases like Expression Atlas. The proteomics community (PRIDE datasets) in contrast to transcriptomics tends to reanalyse the data within 3 years of its publication. Typically, the number of reanalyses in OmicsDI grows within the first 5 years making this a metric better suited to measure immediate impact.

The second metric is the number of direct citations in publications for each dataset as previously suggested[14]. The number of datasets with at least one citation in EuropePMC is 58,054 (Table 1). Figure 2b shows the distribution of dataset direct citations by omics type. Transcriptomics datasets are the most cited ones, followed by genomics and multiomics datasets. Interestingly, the standard deviation indicates that in transcriptomics some datasets get significantly more attention from the community than others (STD = 16), whereas for proteomics datasets the citation rate is much more homogenous (STD = 1.7). The current workflow searches EuropePMC using all the identifiers associated with a given dataset (e.g. a given dataset can be cited in a publication using the ArrayExpress, GEO or BioProject identifiers). For example, the dataset E-GEOD-2034 (https://www.omicsdi.org/dataset/arrayexpress-repository/E-GEOD-2034) is cited 312 and 28 times, using the ArrayExpress (E-GEOD-2034) and GEO (GSE2034) identifiers, respectively.

| omics type | Number of citations | Number of cited datasets | Number of reanalyses | Number of reanalysed datasets | Number of downloads | Number of downloaded datasets | Number of views | Number of viewed datasets | Number of connections | Number of datasets with connections |
|---|---|---|---|---|---|---|---|---|---|---|
| Genomics | 8152 | 3389 | 1103 | 872 | 1,210,799 | 54,336 | 1,233,388 | 13,441 | 1,041,407,105 | 313,549 |
| Metabolomics | 827 | 117 | - | - | 49,907 | 321 | 253,428 | 2726 | 340,483 | 1340 |
| Models | 3 | 3 | 7190 | 239 | - | - | 435,859 | 7262 | 12,880,012 | 7200 |
| Multiomics | 9911 | 2053 | 5013 | 2422 | 179,669 | 2694 | 860,092 | 7848 | 16,453,633 | 7849 |
| Proteomics | 4624 | 1793 | 3344 | 567 | 153,548 | 5392 | 1,417,107 | 13,015 | 51,857,985 | 20,577 |
| Transcriptomics | 665,022 | 50,699 | 10,527 | 8062 | 208,383 | 3675 | 14,793,937 | 119,139 | 27,696,366 | 118,500 |

Table 1 The number of citations, reanalyses, downloads, views and connections (April 2019)

**Biological entity connections**. We analysed the number of biological entities reported on each omics dataset (e.g. UNIPORT proteins) stored in other knowledge-bases (UniProt) (Table 1). More than 53% of the datasets contains biological connections that can be traced to knowledge-based resources, such as Ensembl[15], UniProt[16] or IntAct[17]. The number of connections across different omics types can differ significantly. For example, dataset E-MTAB-599 ("RNA-seq of mouse DBA/2JxC57BL/6J heart, hippocampus, liver, lung, spleen and thymus"), associated with this publication[18], has 1,710,979 connections, including 1,689,177 genome variants, 21,572 gene values and 230 other connections, ranging from sample annotations to nucleotide sequences. The second most connected dataset in the Metabo-Lights database (MTBLS392—https://www.omicsdi.org/dataset/metabolights_dataset/MTBLS392), associated with this publication[19], only contains 345 metabolites reported in the ChEBI database[20]. To overcome these differences, we have implemented a normalisation method that creates a connectivity score which boosts how much a dataset contributes to a specific knowledge-base and also boost datasets that are included in more knowledge-bases (Supplementary Note 1).

We have studied the correlation between all the metrics for the different omics types (Fig. 3). The number of reanalyses and citations are highly correlated for proteomics datasets ($R = 0.7$) but are not correlated for other omics fields, such as transcriptomics, genomics and multiomics: 0.018, 0.02 and 0.12, respectively. The highest global correlation ($R = 0.5$) is observed for the combination of number of connections and downloads. Generally, the five metrics are not correlated for any of the omics fields (Fig. 3) and can be seen as orthogonal variables to get a broader representation of the impact of omics datasets.

## Discussion

One of the obstacles to achieving a systematic deposition of datasets in public repositories is the lack of a broad scientific reward system, considering other research products in addition to scientific publications[7]. Different studies have demonstrated the need for metrics and frameworks to quantify the impact of deposited datasets in the public domain. Such a system would not only encourage authors to make their data public, but also would help funding agencies, biological resources and the scientific community as a whole to focus on the most impactful datasets. In OmicsDI we have implemented a novel platform to quantify the impact of public datasets systematically, by using data from biological data resources (reanalyses), literature (citations), knowledge bases (connections), views and downloads. Every metric is updated on a weekly basis and made available through the OmicsDI web interface and API.

One of the primary findings is that in systems biology (the BioModels database[21] is the representative resource), the deposition of data has enabled systematic generation of new knowledge (biological models) based on previous datasets. For example, the model "Genome-scale metabolic modelling of hepatocytes reveals serine deficiency in patients with non-alcoholic fatty liver disease" (MODEL1402200003)[22] has been used to build more than 6000 models available in BioModels. We noticed different complex graph patterns of reanalysis in the BioModels database. For example, Fig. 1a shows the reanalysis network of model BIOMD0000000055, where the original model published in 2006 has been reused to build new models until 2015. BioModels can be built from multiple models and originated new models (Fig. 1b). BioModels database has defined during the submission process a mechanism to annotate if the model reuse parts of previously published models enabling OmicsDI to build and trace reanalysis patterns. In contrast to

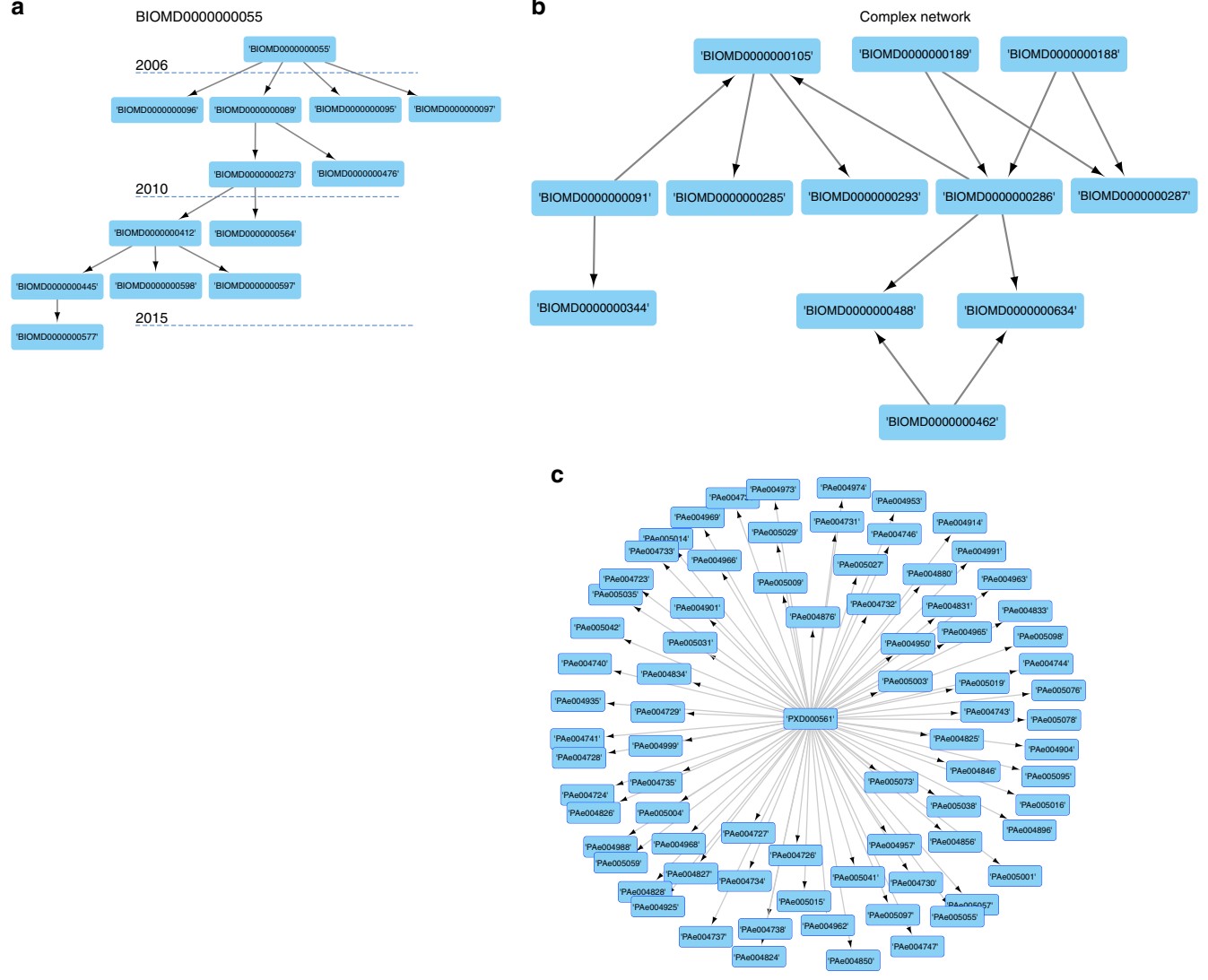

**Fig. 1** Examples of the reanalysis network for different OmicsDI datasets: **a** BioModels model BIOMD0000000055. BioModels are reused over time (e.g. 2006–2015) to build new models; in the BioModel database each new model contains references to the original source model of information. **b** Twelve different BioModels models are connected through a reanalysis network. The BioModel database traces the origin of each model and the relations between them, enabling to trace complex reanalysis relations where models can be originated from multiple models and be used by other models. **c** Proteomics reanalysis network for the draft of the human proteome project (PRIDE accession PXD000561). In proteomics, the predominant reanalysis pattern is "one to many", where original deposited submissions are reanalysed in multiple datasets by multiple authors

biological models, the proteomics (Fig. 1c) and transcriptomics fields are still working to define a proper mechanism to report the multiple reanalyses of datasets in a hierarchical manner[11]. For this reason, the reanalysis pattern detected in proteomics are "one to many" networks where one dataset has been reanalysed by multiple datasets (e.g. PXD000561).

Moreover, the results showed that the reanalysis metric is crucial to highlight relevant datasets early after the dataset release (Fig. 2c). Overall, 8000 datasets (>5% of OmicsDI content) have been reanalysed by resources, such as PeptideAtlas, GPMDB or Expression Atlas, among others. However, it should be noted that the reanalysis metric measures only the impact of datasets in the same or in other data resources contributing their metadata to OmicsDI, which constitutes a fraction of the total re-use by the scientific community.

To complement the reanalysis metric, we counted direct citations of datasets in scientific publications. Different studies have estimated that the proportion of the total citation count contributed by data depositions is around 6–20%[10,14]. Most of the

reanalyses tracked in OmicsDI have been performed using GEO datasets, which might have biased the results to a specific resource. However, our findings show the same patterns in the literature: almost 9000 datasets have been cited in publications at least once. It is important to highlight that counting direct database citations in the whole text of manuscripts is only possible for open access publications. In the case where the corresponding publications are not open access, dataset identifiers would need to be included in the PubMed abstract to be included in this metric. The coverage of direct citations in publications is therefore limited by this systemic issue. We have found that the transcriptomics community (individual researchers) tend to cite the same datasets more often, with an average of four citations per dataset. The most cited dataset is "Transcription profiling of human breast cancer samples—relapse free survival" (E-GEOD-2034), totalling 312 citations. Both metrics, reanalyses and citations, should be used in combination for a better understanding of the dataset impact. Our results show that both metrics are uncorrelated and should not be aggregated. For example, dataset

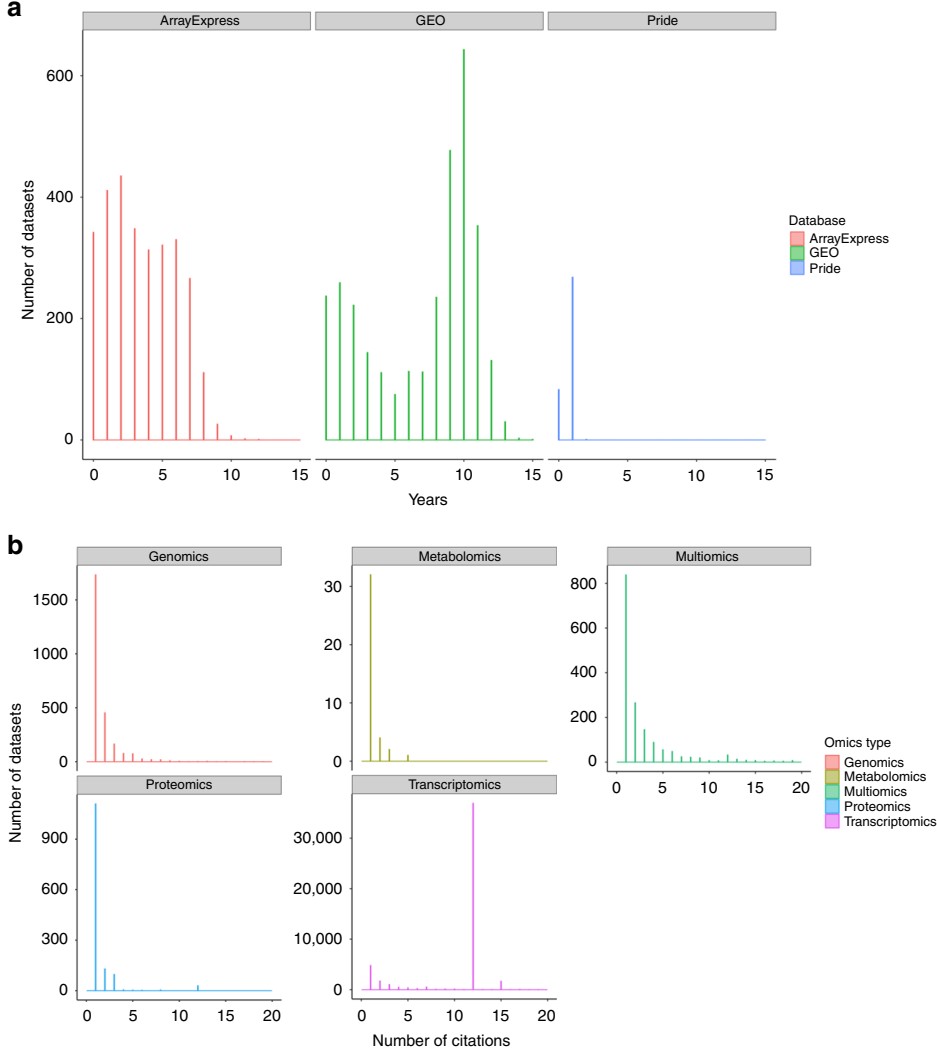

**Fig. 2 a** Elapsed time between the original publication of a dataset and the publication of all its reanalyses for three omics data archives (PRIDE—Proteomics, GEO—Transcriptomics, ArrayExpress—Transcriptomics). Transcriptomics datasets tend to be reanalysed over time until datasets are 12 years old, while proteomics datasets (PRIDE) are less reused after 3 years from their publication. **b** Distribution of the number of citations per dataset group by OmicsDI omics type. Transcriptomics datasets are highly cited with more than 30,000 datasets with 11 citations; while in genomics, proteomics and metabolomics most datasets are only cited once

E-MTAB-513 is among the 10 most cited datasets in the literature: it has been cited 155 times and reanalysed 4 times. In addition to the normalised values, we have decided to provide the "raw" metrics to the community, which will enable to combine them into more complex models[23]. However, we have shown that these metrics can be used independently to generate models for clustering and classification (Supplementary Note 2).

In 2011, Mons et al. introduced the idea of nano-publications, from which the authors could get credit not only through the actual publication but also through all the knowledge associated with it[7]. In our view, the value of the dataset should not be only associated with the "raw data" or the claims in the publication, but also should be assessed considering all the biological entities supported in knowledgebases. We have developed the connections metric, which can be used to estimate the impact of a dataset for knowledgebases, by counting how many biological entities are supported by it.

Importantly, OmicsDI is monitoring not only the web interface views but also the interaction through the OmicsDI API. On average, every dataset in OmicsDI has been accessed at least 30 times since 2016 (Table 1). By March 2019, we had captured the number of direct downloads for six different databases at the European Bioinformatics Institute. These two metrics (views and downloads) are not publicly available in any of these resources and at present are infeasible to retrieve. In fact, at present, the first coordinated efforts to gather them in a standard manner are taking place in the context of the ELIXIR framework for European biological data resources[24]. With this first implementation, we are promoting that resources systematically release this information to the public domain.

The newly implemented OmicsDI dataset claiming system enables authors, research groups, scientific consortia and research institutions to organise datasets under a unique OmicsDI profile, and for datasets to be added to their own ORCID profiles as well. At the time of writing (March 2019), 968 datasets have been claimed into ORCID profiles through OmicsDI. In our view, following the same system for monitoring the impact of individual datasets, these metrics could also be used to measure at least some aspects of the impact of public omics data resources[25,26]. A common problem of impact evaluation is to compare different fields or topics with the same metrics. Figure 4 shows the average distribution of metrics (raw and normalised) for each omics type.

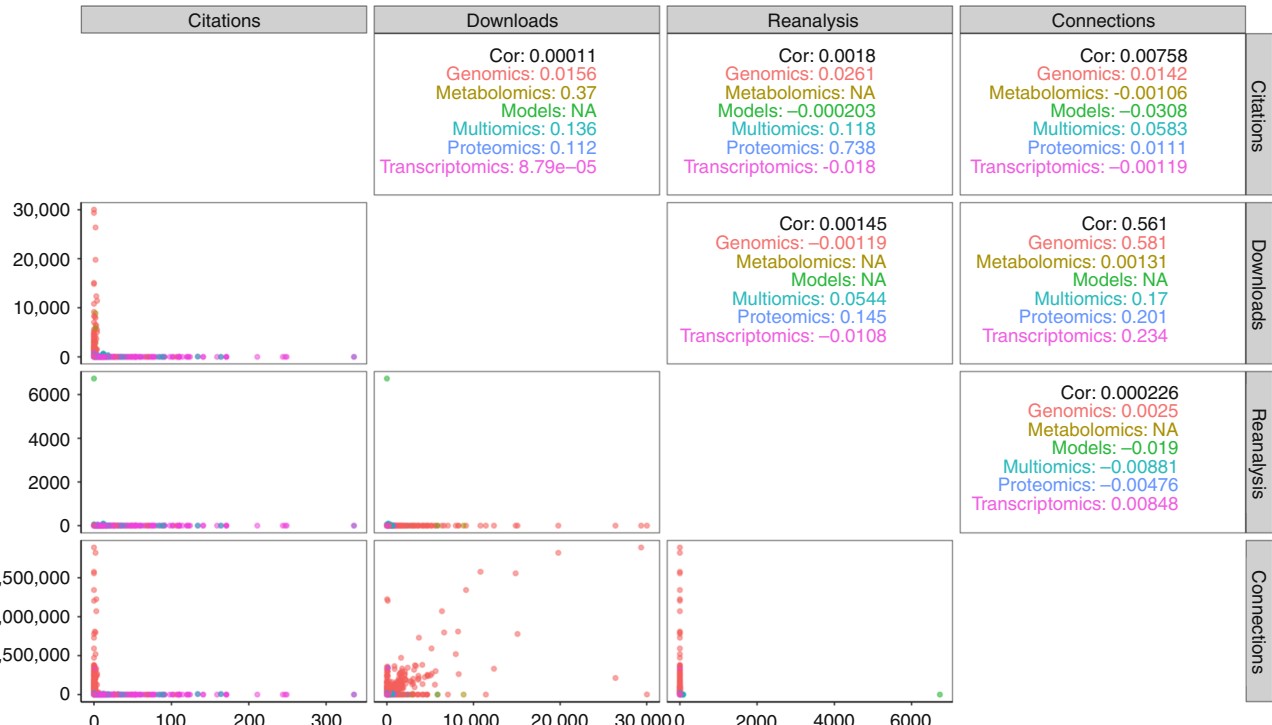

**Fig. 3** Correlation between the OmicsDI metrics (reanalyses, citations, downloads and connections) and omics type (proteomics, genomics, transcriptomics, metabolomics and multiomics). The lower left panels show the scatter plots of each combination and the upper right panels shows the values for each type of omics. For example, the number of downloads and the number of connections of Genomics datasets (red) is highly correlated ($R = 0.5$) compared to all other metrics combinations

We observed major differences across omics types, which demonstrate that each field has different patterns for each metric. Therefore, we recommend the use of these metrics to evaluate datasets within the same omics field, as classified by OmicsDI[1].

Several studies have discussed the challenges of collecting impact metrics for manuscripts and datasets[14,27,28]. With the developments of new services such as the EuropePMC API, the compilation of direct citations for datasets has become more feasible[7]. However, in our view, some conventions should be implemented to normalise the way datasets are cited:

i.  When the dataset is the main focus of work, dataset identifiers should be used, instead of citing the corresponding publications as suggested by the FORCE11 group[29].
ii. The scientific community needs to develop standard citation strategies for datasets. For example, ~60% of the data re-used in one of the drafts of the human proteome[27] was collected from public repositories. However, no proper references to the original authors and data were present in the main text of their manuscripts. In order to be able to properly cite the re-use of datasets, new mechanisms should be developed to enable an adequate reporting. OmicsDI has implemented a visualisation component (Supplementary Note 3) that allows users to cite the corresponding dataset using the FORCE11 Data Citation Synthesis Group.
iii. Repositories should make openly available (in an easy to retrieve manner) the links between their reanalysed and the original datasets. Good examples of these links can be found already in Expression Atlas and PeptideAtlas, where every reanalysed dataset references the original ones (Supplementary Note 4). Indeed, many databases reference only the associated publications rather than the actual dataset identifiers. The correct tracking of datasets in a database by other data resources can help to assess its impact, since it

demonstrates that the data they store is actively re-used by (and thus it is relevant to) the community. Naturally, the same effort should also be made by knowledgebases (e.g. resources including pathways, interactions, gene/protein profiles, etc.), to reference the original datasets rather than the publications, in order to recognise that a large part of the biological knowledge is derived from the actual datasets.

We envision that as more and more data is made publicly available, more standardisation will be implemented to cross-link resources, manuscripts, datasets and the final biological molecules, making the proposed framework more robust. We expect that any mature omics field should welcome novel insights that can be derived from existing datasets and promote their traceability. We all "stand on the shoulders of giants". Therefore, we expect that an improved quantitation of the impact of datasets will help scientists, funders and research organisations to better value a broader range of "giants" (research products).

## Methods

**Omics datasets impact metrics**. In contrast to publications, where the impact is mainly measured by the number of citations, we believe the impact of datasets should be quantified using more than one metric. We have formulated five metrics that can be used to estimate the impact of datasets (Fig. 5):

1.  *Number of reanalyses (reanalyses)*: A reanalysis can be generally defined as the complete or partial re-use of an original dataset (A) using a different analysis protocol and stored either in the same or in another public data resource (B) (Fig. 5). For example, PeptideAtlas[30] systematically reanalyses public proteomics datasets, mainly from PRIDE[31] and MassIVE (massive.ucsd.edu). The new peptide and protein evidences from these reanalyses have become an invaluable resource, e.g. for the Human Proteome Project[12]. The appropriate and accurate reference to the original datasets in other resources facilitates the reproducibility and traceability of the results and the recognition for the authors that generated the original dataset[32]. We implemented the *reanalysis* metric using the OmicsDI XML schema (https://github.com/OmicsDI/specifications), which provides a mechanism to define when a dataset is based on another one (Fig. 5).

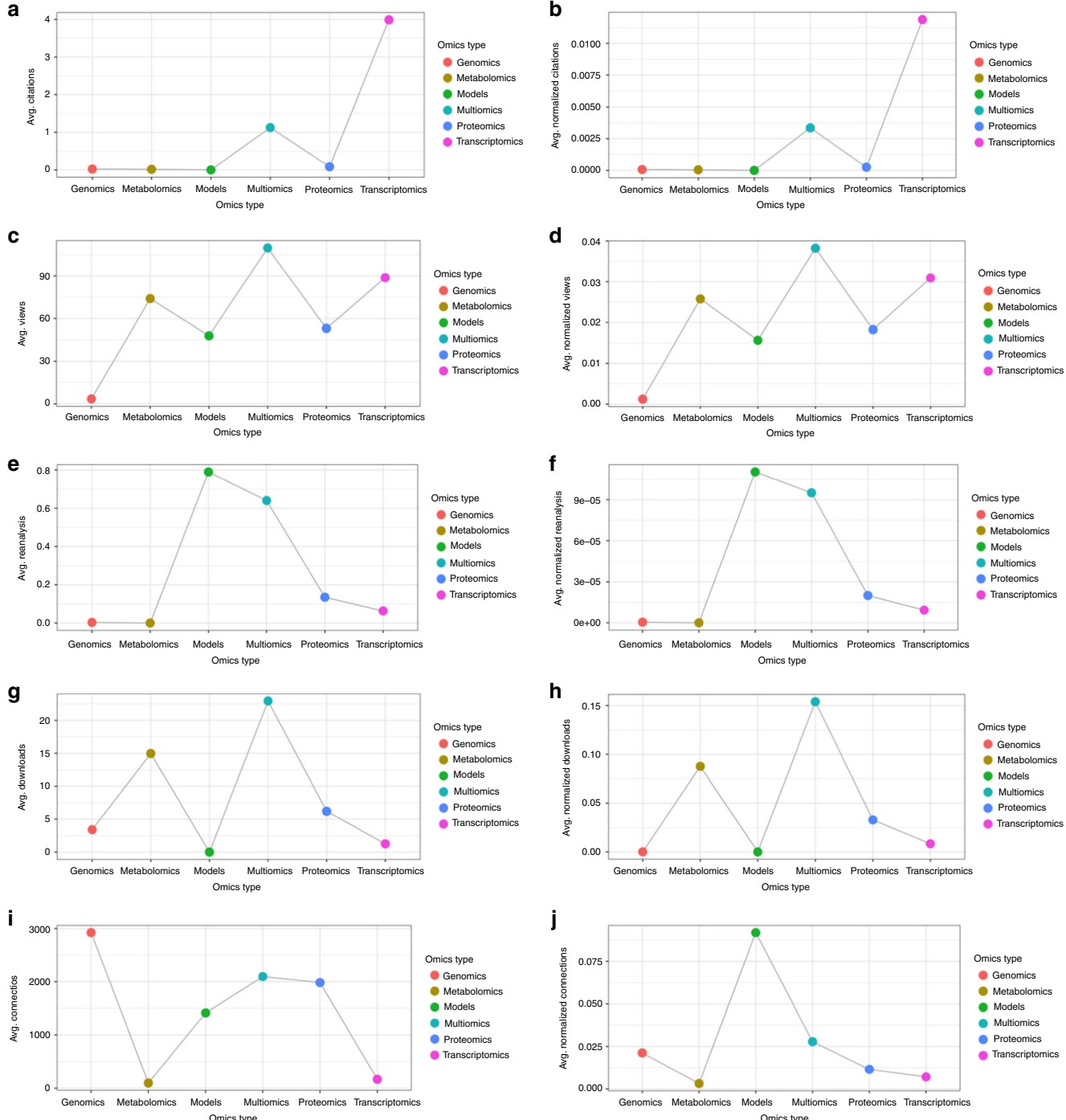

**Fig. 4** Average distribution of each metric (*citations, views, reanalysis, downloads* and *connections*) by omics type: raw (**a**, **c**, **e**, **g**, **i**) and normalised values (**b**, **d**, **f**, **h**, **j**). The raw values are the metrics values collected with the OmicsDI pipelines, whereas the normalised values are the transformation of those values using MinMax scaler or the Biological connections normalisation method

2. *Direct citations of dataset identifiers in publications (citations)*: Citations represent the number of publications that directly refer to dataset identifiers[14]. We use the EuropePMC API (europepmc.org/RestfulWebService)[33] to query all the open access publications that directly cite a dataset accession. EuropePMC stores scientific publications (and also pre-prints) that are open access, so the whole text of the manuscript can be used for performing text mining, enabling this functionality. With increasing standardisation of data citations in the scientific literature, as proposed by Fenner et al. [29], this metric is expected to become easier to systematically generate in the future.

3. *Number of downloads and views (downloads and views)*: The number of views and downloads of each dataset can be used to estimate the number of times a dataset is used even if it does not get cited or publicly reanalysed. These two metrics have recently been proposed by different journals

(e.g. PLOS https://www.plos.org/article-level-metrics) to complement the number of manuscript citations. By 2019, we were able to systematically trace the number of direct downloads from six different data resources (PRIDE, ArrayExpress, MetaboLights, EVA, Expression Atlas and ENA). Additionally, we provide counts of the number of views and accesses for each dataset coming from all database providers via the OmicsDI web and restful API (https://www.omicsdi.org/ws/), respectively.

4. *Number of biological entities claims based on the dataset (connections)*: Here, we provide counts of the number of biological entities which are reported by a given dataset, in various popular biomedical knowledge bases, such as UniProt (protein sequences), IntAct[34] (molecular interaction data) or Reactome[35] (biological pathways) (Fig. 5b), (Supplementary Note 2). Most of the omics datasets support biological claims (e.g. pathways, interactions, expression

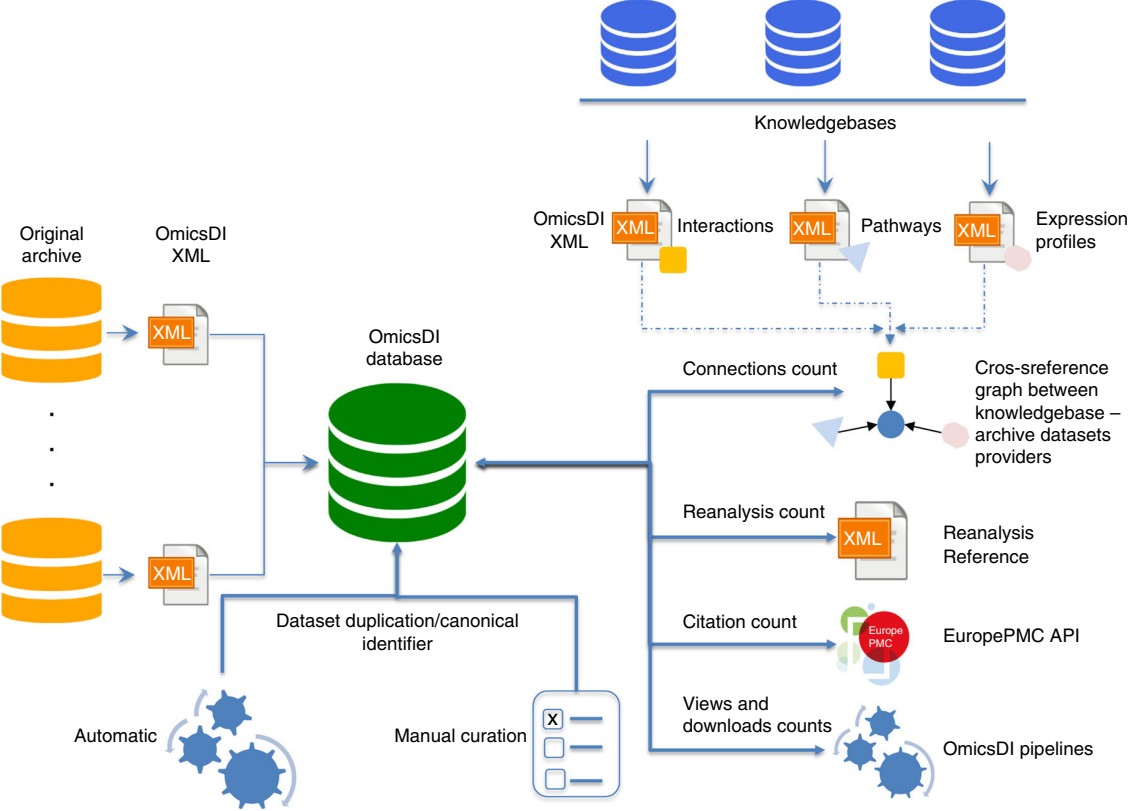

**Fig. 5** The metrics estimation pipeline is based on the OmicsDI XML file format that is used to transfer datasets from each provider into OmicsDI. Data is imported from each provider into a central MongoDB database. An automatic pipeline is run to detect duplication and data replication across the resource. The pipelines use the central MongoDB database, the EuropePMC API and the knowledgebases (e.g. Ensembl, UniProt) to compute/estimate the different metrics

profiles, etc.) that are either manually curated or automatically annotated in relevant knowledge bases. We believe that calculating impact in this manner is needed, since for instance, if a number of gene expression profiles were supported by one particular dataset, that information contribution would be lost (or at least untraceable) if that dataset was no longer publicly available. All biological knowledgebases are indexed and queried by OmicsDI using the EBI Search indexing system (Supplementary Note 1).

**Metrics normalisation**. We have implemented two different normalisation methods to adjust all the metrics generated on different scales to a notionally common scale. For citations, views, downloads and reanalyses metrics the normalisation of each value was estimated using the MinMax Scaler method[36,37]. The MinMaxScaler is a robust method to shrink original values of a distribution to a range such that it becomes a value between 0 and 1. This normalisation method works better for cases in which the values distribution does not follow a Gaussian model and the standard deviation is relatively small[36,37]. However, for the connection metric a different normalisation method needed to be applied because depending on the omics type the number of reported biological entities could vary widely. For example, a dataset within the genomics type can contain millions of connections (PRJNA391943 in ENA has 9,443,069 genomic sequence variant sites) and the datasets with most connections in metabolomics can contain only 3113 connections (i.e. metabolites).

OmicsDI connection normalisation (connectivity score) measures the average ranking of a dataset across all knowledgebases. There are five ranking levels (corresponding to the 20, 40, 60, 80 and 100 percentiles of connections). We first calculate which percentile a dataset falls into, in any given domain or knowledgebase. Datasets in a 0–20 percentile get rank 1, in a 20–40 percentile get rank 2 and so on. Then, the final score is calculated by averaging these ranks across all knowledgebases. This normalisation method boosts datasets that are included and referenced by multiple knowledgebases. It also ensures that as OmicsDI grows, by including more datasets and omics domains, the score will become more robust against outliers in the future.

**Proposed platform to quantify dataset impact**. In order to compute the proposed metrics (reanalyses, citations, views, downloads and connections), a novel platform has been developed within OmicsDI. OmicsDI imports the metadata for

each dataset from the original providers, using the OmicsDI XML file format (Supplementary Note 5)[1]. In order to ensure that the metrics are accurate, the infrastructure implements a system to remove dataset redundancy (when two different resources store the same original dataset). An automatic pipeline and a manual annotation system enable OmicsDI to group duplicated datasets with potentially different identifiers (e.g. transcriptomics datasets available in ArrayExpress and Gene Expression Omnibus (GEO)) (Fig. 5c). In this case, the pipeline designates one of the datasets as the canonical representation and annotates the rest of identifiers as additional secondary ones. All these pipelines and software components can scale to handle thousands of datasets and systematically compute and update the metrics on a weekly basis (github.com/OmicsDI/index-pipeline).

The current OmicsDI web interface includes for each dataset a new badge (Rosette flower) including all the metrics proposed (Fig. 6). The number in the centre of the Rosette's shows the aggregated score (Omics score). The Omics score is computed using all the normalised metrics (reanalyses, citations, views, downloads and connections). The formula for this metric will be estimated as described:

$$\sigma_{d \in \mathrm{Odi}} = \left( \frac{Co_d}{\sum Co_{\mathrm{Odi}}} + \frac{Ci_d}{\sum Ci_{\mathrm{Odi}}} + \frac{P_d}{\sum P_{\mathrm{Odi}}} + \frac{V_d}{\sum V_{\mathrm{Odi}}} + \frac{D_d}{\sum D_{\mathrm{Odi}}} \right) \times 1000$$

The intensity of the colour leaf in the flower will change depending on the value of the metric with respect to all datasets.

**Dataset claiming component**. Analogously to services such as Google Scholar and ResearchGate for publications, we have implemented a mechanism that enables researchers to create their own profile in OmicsDI, by claiming their own datasets. Researchers need to log into OmicsDI using their corresponding ORCID account details (Supplementary Note 7), and search for relevant datasets using different criteria such as: (i) dataset identifiers, (ii) specific keywords in the title or description of the dataset, or analogous information from the corresponding manuscript where the generation of the dataset is reported, and/or (iii) the author's name, among others. Then, datasets can be added to an OmicsDI personal profile where it is possible to visualise the impact metrics (reanalyses, citations, views, downloads and connections), providing to researchers this rich information. The URLs of personal profiles can be shared with anyone in the community. Additionally, as a key point, OmicsDI claimed datasets can be synchronised to the researcher's own ORCID profile, highlighting datasets there as a research product

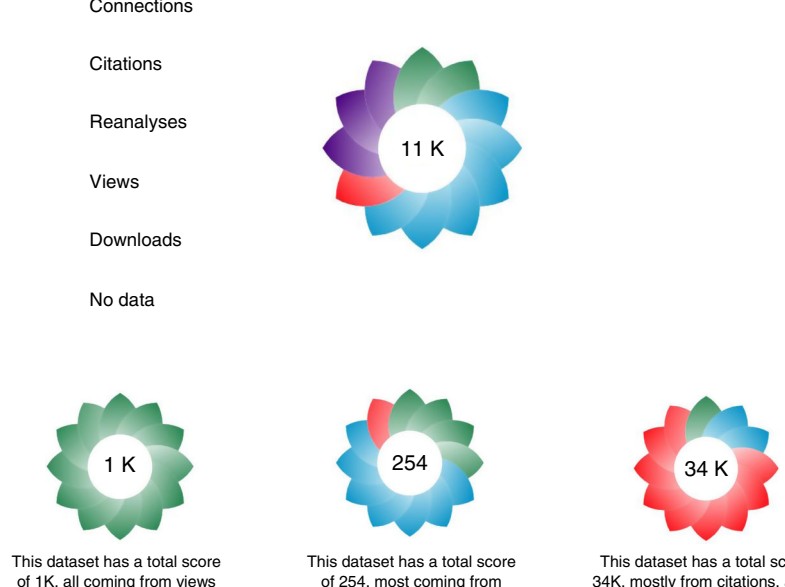

The colours of the rosette flower each represent a different type of dataset metric:

Connections

Citations

Reanalyses

Views

Downloads

No data

11 K

1 K

This dataset has a total score of 1K, all coming from views

254

This dataset has a total score of 254, most coming from connections. around 30% comes from views and a small part comes from citations

34 K

This dataset has a total score of 34K, mostly from citations. around 15% comes from connections and 7% comes from views

**Fig. 6** The OmicsDI badge (Rosette flower) represents all the OmicsDI metrics. In the centre of the badge the OmicsDI score estimates the global impact using all metrics

as well[38]. Although this mechanism is initially aimed at individual researchers (e.g. https://www.omicsdi.org/profile/xQuOBTAW), research groups, scientific consortia (e.g. https://www.omicsdi.org/profile/ZEd3mwfF), and research institutions can also create their own OmicsDI profile, facilitating the aggregation, visualisation, tracking and impact assessment for their generated datasets and the addition to their own OmicsDI profiles. In addition, OmicsDI has implemented a simple visualisation component that allows users to cite the corresponding dataset using the FORCE11 Data Citation Synthesis Group recommendations (http://www.dcc.ac.uk/resources/how-guides/cite-datasets).

**Reporting summary**. Further information on research design is available in the Nature Research Reporting Summary linked to this article.

## Data availability
The datasets mentioned in the current study are available in BioModels, ArrayExpress and PRIDE databases. No datasets were generated during the current study.

## Code availability
All code supporting the current study is deposited in GitHub under the organisation (https://github.com/OmicsDI).

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

## Acknowledgements

Y.P.-R. and A.Z. were supported by US NIH BD2K grant U54 GM114833. G.D. is supported by EMBL core funding. P.X. was supported by BBSRC International Part-nering Award BB/N022432/1. Y.P.-R. and J.A.V. acknowledge the Wellcome Trust (grant number 208391/Z/17/Z) and the EPIC-XS project (grant number 823839), funded by the Horizon 2020 programme of the European Union.

## Author contributions

Y.P.-R. conceived the idea and designed the resource. H.H. supervised the project. The OmicsDI platform was implemented by Y.P.-R., A.Z., G.D., M.-T.V, P.X. R.P. implemented the download metrics component. A.F.J. developed the statistics for the biological connection metrics. M.G. contributed to the design of the reanalysis concept for model data. J.A.V. and P.P. made complementary contributions and provided feedback on the resource design based on their respective backgrounds. Y.P.-R. and H.H. wrote the manuscript with the assistance and feedback of all other co-authors.

## Additional information

**Competing interests:** The authors declare no competing interests.

