## [Peer Review File · Nature Communications]

Editorial Note: This manuscript has been previously reviewed at another journal that is not operating a transparent peer review scheme. This document only contains reviewer comments and rebuttal letters for versions considered at Nature Communications . Mentions of prior referee reports have been redacted.

REVIEWERS' COMMENTS:

Reviewer #2 (Remarks to the Author):

The revised manuscript regarding updates to the Omics Discovery Index has important improvements compared to the prior version of the manuscript. The development and inclusion of OmicsDI score is a strong addition to the manuscript and is a good starting point for future considerations regarding how to evaluate omics datasets. This manuscript is appropriate for Nature Communications after carrying out important revisions.

The main issue is with the figures and figure legends associated with the main body of the manuscript. Each of the five figure legends are far too brief and do a very poor job describing the content of each figure. Following standard practice each figure should have a title followed by a several sentence detailed description of the figure. Figures 2-4 are also problematic in general with small fonts and difficult to discern color schemes, for example.

Figure 2 contains three different types of reanalysis, which is interesting, but needs reconsideration as a presentation style and careful description of what each panel represents in the legend. Figure 3 has odd positioning of both panels and again suffers from small fonts.

Figure 3 would be enhanced also with panels that show the number of reanalyses of selected individual datasets from year to year. It would be interesting to see if there are differences in times to reanalysis for particular fields and or datasets. Are there certain datasets that are far ahead of their time and only get reanalyzed several years later?

Figure 4 essentially has 25 panels and is very difficult to interpret especially considering the very sparse figure legend. It is unclear what is in each panel, why the information is present, and what the figure is trying to convey. The description of Figure 4 on page 8 is also quite brief considering the number of panels in the figure. The authors need to carefully consider what this figure is trying to convey and rewrite the description of this figure providing more details and referring to specific panels to highlight specific results.

Finally, as a minor revision, figure 1 in the supplement is a good figure and helps frame the manuscript. It would be valuable to move this to the main body of the manuscript and then reorder/renumber the current figures.

Reviewer #2 (Remarks to the Author):

The main issue is with the figures and figure legends associated with the main body of the manuscript. Each of the five figure legends are far too brief and do a very poor job describing the content of each figure. Following standard practice each figure should have a title followed by a several sentence detailed descriptions of the figure. Figures 2-4 are also problematic in general with small fonts and difficult to discern color schemes, for example.

R/ Thanks to the reviewer for this comment. We didn't add more information into the legend of the figure because we thought they were self-explained. We agree with the reviewer and several sentences has been added to the legends of all figures. In addition, we have reviewed the font and resolution of all figures.

Figure 2 contains three different types of reanalysis, which is interesting, but needs reconsideration as a presentation style and careful description of what each panel represents in the legend. Figure 3 has odd positioning of both panels and again suffers from small fonts.

R/ We have added in the legend a few sentences to explain each reanalysis pattern. In addition, we have corrected the quality of the figure and the font size. We removed the one of the proteomics datasets because the reanalysis pattern is similar to the previous one.

Figure 3 would be enhanced also with panels that show the number of reanalyses of selected individual datasets from year to year. It would be interesting to see if there are differences in times to reanalysis for particular fields and or datasets. Are there certain datasets that are far ahead of their time and only get reanalyzed several years later?

R/ The main idea we want to discuss with this plot is that datasets in Transcriptomics get reanalyzed even when they are more than 10 years old. We want to make clear also that datasets get attention and impact more quickly than citations to the corresponding manuscript. We also want to motivate the fields of proteomics and metabolomics to reuse more data. The current plot has been separated into three different databases GEO, ArrayExpress and PRIDE to make clearer the trends for different omics fields.

Figure 4 essentially has 25 panels and is very difficult to interpret especially considering the very sparse figure legend. It is unclear what is in each panel, why the information is present, and what the figure is trying to convey. The description of Figure 4 on page 8 is also quite brief considering the number of panels in the figure. The authors need to carefully consider what this figure is trying to convey and rewrite the description of this figure providing more details and referring to specific panels to highlight specific results.

R/ We agree with the reviewer and in the present version we have removed some of the panels and only leave the important results. Also, the legend contains more details about the panels.

Finally, as a minor revision, figure 1 in the supplement is a good figure and helps frame the manuscript. It would be valuable to move this to the main body of the manuscript and then reorder/renumber the current figures.

R/ We had the architecture/workflow figure in the main manuscript in the first review iteration but one of the reviewers suggested to move it into Supplementary Information. We appreciated this suggestion and think that the figure should be in the main text to support the claims and explain how we compute each metric.